# Hepatocellular carcinoma presentation and prognosis among Nigerian adults with and without HIV

**Pantong M. Davwar**[1], **Edith Okeke**[1], **Mary Duguru**[1], **David Nyam**[1], **Kristen Bell**[2], **Emuobor A. Odeghe**[3], **Ganiat Oyeleke**[3], **Olufunmilayo A. Lesi**[3], **Revika Singh**[2], **Kwang-Youn Kim**[2], **Godwin Imade**[1], **Alani S. Akanmu**[3], **Atiene S. Sagay**[1], **Folasade T. Ogunsola**[3], **Marion G. Peters**[2], **Lewis R. Roberts**[4], **Lifang Hou**[2], **Robert L. Murphy**[2], **Claudia A. Hawkins**[2]*

1 University of Jos, Jos, Nigeria, 2 Feinberg School of Medicine - Northwestern University, Chicago, Illinois, United States of America, 3 University of Lagos, Lagos, Nigeria, 4 Mayo Clinic College of Medicine and Science, Rochester, Minnesota, United States of America

* c-hawkins@northwestern.edu

## Abstract

### Introduction

Hepatocellular carcinoma (HCC) is an increasing cause of mortality in Nigeria among persons with HIV (PLH), as access to antiretroviral therapy (ART) improves. In this study we describe clinical, radiological, and laboratory characteristics in Nigerian adults with HCC, with and without HIV, and examine how HIV impacts survival.

### Methods

This prospective observational study was conducted between August 2018 and November 2021 at two Nigerian hospitals [Jos University Teaching Hospital (JUTH) and Lagos University Teaching Hospital (LUTH)]. Subjects ≥18 years with HCC diagnosed according to American Association for the Study of Liver Diseases (AASLD) criteria were included. Baseline characteristics were compared, and Kaplan-Meier curves were generated to estimate survival.

### Results

213 subjects [177 (83%) without HIV and 36 (17%) with HIV (PLH)] were enrolled. Median age was 52 years (IQR 42,60) and most subjects were male (71%). 83% PLH were on antiretroviral therapy (ART). Hepatitis B surface antigen (HBsAg) positivity was similar between the two groups [91/177 (51%) without HIV vs. 18/36 (50%) with HIV; p = 0.86]. 46/213 (22%) subjects had active hepatitis C (anti-HCV+/HCV RNA>10 IU/mL). Cirrhosis was more common in PLH but there were no other significant differences in clinical and tumor characteristics between the groups. Overall, 99% subjects were symptomatic and 78% in late-stage HCC. Median overall survival was significantly shorter in PLH vs. without HIV (0.98 months vs 3.02 months, HR = 1.55, 95%CI 1.02, 2.37, p = 0.04). This association was not significant

data are available from the Institutional Review Board at Northwestern University and the University of Jos for researchers who meet the criteria for access to confidential data. Contact details for the NU IRB are Name: Braden Van Buskirk, Address:Arthur Rubloff Building, 7th Floor 750 N. Lake Shore Dr. Chicago, IL 60611 312-503-9338.

**Funding:** Division of Cancer Prevention, National Cancer Institute. Grant Number: NIH/NCI U54CA221205 (PI Robert Murphy, Lifting Hou) The funders had no role in study design, data collection and analysis, decision to publish, or preparation of the manuscript.

**Competing interests:** The authors have declared that no competing interests exist.

after adjusting for known risk factors including gender, current alcohol use, alpha-fetoprotein (AFP), albumin, and total bilirubin (HR = 1.38, 95%CI 0.84, 2.29, p = 0.21).

## Conclusion

HCC presented late with an extremely poor overall prognosis, highlighting the urgent need for more intensive surveillance in Nigeria to diagnose HCC at earlier stages. Early diagnosis and management of viral hepatitis, and access to HCC therapies, could prevent early mortality among persons with HCC, especially among PLH.

## Introduction

Hepatocellular carcinoma (HCC) is the sixth most-common cancer and the third most-common cause of cancer-related mortality in the world [1]. HCC is a major public health problem in West Africa. In Nigeria, HCC incidence is 8.4/100,000 persons and has a nearly identical mortality rate because most people present with late-stage disease due to a lack of adequate surveillance [1, 2].

HCC is emerging as an increasing cause of mortality in persons with HIV (PLH) with increased access to antiretroviral therapy (ART) [3, 4]. In the US, PLH have been shown to have a 3–7 times higher incidence of HCC compared to those without HIV [5]. HCC incidence in PLH in West Africa has not been well characterized, despite this region having some of the highest HCC and HIV rates in the world. In Nigeria, approximately 1.9 million persons are living with HIV [6]. PLH in Nigeria also have a high prevalence of co-infection with HBV (7.0–10.0%) and HCV (1.6–6.5%), both well-known risk factors for HCC [7–10]. HBV has been found to be present in 50% of persons with HCC [11]. High rates of HCV antibody seropositivity among Nigerians with HCC have also been reported, although rates of chronic HCV infection (detectable HCV viral load) are unknown [12]. Locally, many other factors exist that substantially increase the risk of HCC including exposure to aflatoxin, use of herbal medication, dietary iron overload, excess alcohol consumption and fatty liver disease, some of which are also more prevalent in PLH [13].

In addition to the increased burden of HCC, several studies have also reported faster progression of HCC and high mortality rates in PLH, particularly those with viral hepatitis co-infection [14]. In Sub-Saharan Africa (SSA) where a high burden of HIV, viral co-infections and HCC exists, there is a need to further elucidate the impact of HIV and other viral co-infections on HCC progression and survival and consider whether more aggressive surveillance strategies are needed in PLH. In this study, we describe the clinical, radiological, and laboratory features of HCC in Nigerian adults with and without HIV, and compare median survival in HCC between the two groups. Understanding these differences could better help inform treatment and surveillance strategies that could improve morbidity and mortality from HCC in this high-risk population.

## Methods

This prospective study included a cohort of adults with newly diagnosed HCC with and without HIV enrolled in an NCI-funded study examining epigenomic biomarkers associated with HIV-associated HCC (Award number U54CA221205). Subjects with HCC were recruited between August 2018 and November 2021 from the Hepatology clinics and inpatients wards

of both the Jos University Teaching Hospital (JUTH) and Lagos University Teaching Hospitals (LUTH), and their affiliated sites in Nigeria.

Inclusion criteria were: a) ≥18yrs; b) confirmed to have HCC diagnosed by ultrasound and computed tomography (CT) scan. Exclusion criteria were: a) other known primary or secondary malignancy either currently or within the past 5 years. HCC was diagnosed based on radiologic criteria using triple phase CT according to the AASLD 2018 guidelines [15], which was offered to all patients who had a ≥1cm nodule on ultrasound and/or risk factors for HCC (i.e. HIV, HBV, HCV infection or cirrhosis of any etiology), and/or other symptoms and signs suggestive of HCC. CT findings of arterial phase hyperenhancement and washout during the portal venous phase were considered diagnostic. A liver biopsy was not required for diagnosis, and was not performed in any of the study subjects.

All study subjects confirmed with HCC by CT were provided with information about the study and asked to sign an informed consent. At their enrollment visit, participants underwent a physical exam by the study physician and laboratory testing for HIV (ELISA and Western Blot); complete blood count (CBC), comprehensive chemistry panel including albumin, total bilirubin, aspartate aminotransferase (AST), alanine aminotransferase (ALT), viral serology including hepatitis B surface antigen (HBsAg) and antibody (anti-HBs) and HCV antibody (anti-HCV) using Lumi Quick diagnostics rapid test kits, and alpha-fetoprotein (AFP). Plasma samples were also collected for methylomic analysis. Patients who were confirmed HIV seropositive, also had CD4+ T cell count (flow cytometry (Partec GmbH, Munster, Germany), and HIV RNA Roche Ampliprep TaqMan (Roche Diagnostics Germany; lower limit of detection (LLD) of 20 cp/ml) testing performed and were referred immediately to the HIV care and treatment clinic at their respective institutions for further treatment and care. Confirmatory HBV and HCV viral load testing on stored specimens was performed on all HBsAg and anti-HCV positive participants using the Cepheid GeneXpert® system. Participants received the usual care and treatment for HCC, HIV, viral hepatitis and other health conditions, according to Nigerian national and international guidelines [16, 17]. Sorafenib is offered to eligible patients with advanced disease and Child Pugh Turcotte (CPT) A where available. However, none of the study participants in the study received sorafenib or any other interventional therapies due to lack of access and/or high cost of the medication.

## Data collection

An interviewer-administered questionnaire was used to collect the following demographic, clinical, and tumor data including: age, sex, date of study visit; (ii) BMI, WHO stage (persons with HIV only); (iii) symptoms associated with HCC and/or advanced liver disease i.e. jaundice, abdominal swelling, hematemesis, leg swelling, slow mentation, or confusion; (iv) other medical history; (v) quantification of alcohol intake and history of herbal medicine use; (vi) family history of HCC; (vii) HBV, HCV and HIV treatment history if HBV, HCV and/or HIV positive; (viii) HCC tumor characteristics (from CT) including size of tumor, number of tumors, vascular invasion, presence or absence of ascites, portal vein thrombosis; (xi) antiviral therapies for HBV and HCV, type and duration; (xii) ART therapy and duration (HIV-infected patients on ART); CPT score and Barcelona Clinic Liver Cancer (BCLC) staging were calculated based on available lab, radiologic, and clinical data by a physician.

**Follow up.** All study subjects were followed for 1 year. Subjects' survival was collected monthly via phone calls to participants or participants next of kin or in person if they attended a clinical visit. Date of death was recorded to the nearest possible date if the exact date was unknown. Cause of death was obtained from next of kin or clinical record if they died while hospitalized.

**Ethical approval.** We obtained written informed consent from each participant in this study. The study was approved by the Jos University and Lagos University Teaching Hospitals Ethics Committees and the Northwestern University Institutional Review Board.

## Statistical analysis

Baseline characteristics were summarized using descriptive statistics including median and interquartile range (IQR) for continuous variables, and frequencies and counts for categorical variables. Mann-Whitney U tests were used to compare continuous variables and Fisher's exact tests were used to compare categorical variables. Overall survival (OS) (primary outcome) was defined as time from study enrollment to death with patients censored at date of last available follow-up. To illustrate differences in OS between HIV status groups, Kaplan-Meier curves were presented and differences between groups were compared using the log-rank test. Association between HIV status and OS was assessed using a Cox proportional hazards model before and after adjusting for baseline predictors of interest including gender, current alcohol use, AFP, albumin, and total bilirubin. Results were presented using hazard ratios (HR) and corresponding 95% confidence intervals (CI). Statistical analyses and corresponding figures were generated using R statistical environment (v 4.0.2) along with extension packages survival (v 3.2–13) and survminer (v 0.4.9) [18, 19].

## Results

### Entry characteristics of the study population

213 participants [177 (83%) without HIV, 36 (17%) with HIV; median age (years)52 (IQR = 42, 60); 71% male] were included in this analysis. 109 (52%) were HBsAg positive; 8/109 (7%) were HBeAg positive. HBsAg seropositivity was similar between those with and without HIV. Among HBsAg positives, median HBV DNA was significantly higher in subjects without HIV [36,100 (336–606,750) IU/ml] vs. subjects with HIV [47 (0, 461) IU/ml]; p<0.01. Among subjects with HIV, 30 (83%) were on ART with a median duration of treatment of 8 years (IQR 2,12). ART therapy typically included the anti-HBV active drug tenofovir. 75/213 (36.4%) participants were anti-HCV positive; 46/213 had chronic HCV infection defined as anti-HCV positive with HCV VL >10 IU/L. The proportion with chronic HCV did not differ by HIV status. In participants with HIV, the median CD4+ T cell count at enrollment was 284 (IQR 137, 356) cells/mm$^3$ and 15/28 (54%) with a HIV viral load measurement were HIV virologically suppressed (HIV VL <20 IU/L). Among all participants the prevalence of co-morbidities was low (20%). Almost all (99%) participants presented with ≥1 clinical symptom related to their liver cancer (ie. jaundice, anorexia, weakness, abdominal swelling, abdominal pain, leg swelling, weight loss). 78% presented with a CPT ≥B and 89% with at least one clinical finding consistent with advanced liver disease (ie. ascites, encephalopathy). 78% of participants were classified as BCLC Stage C or D. Staging and clinical symptoms did not differ significantly between persons with and without HIV. The median number of lesions on CT imaging was 6 (IQR 3,11); largest liver mass diameter(cm) was 7.60 (4.7, 11.5) and tumor burden was >50% in 65% of HCC patients. The proportion with cirrhosis detected on CT was significantly higher among subjects without HIV [108 (71%)] vs. those with HIV [13 (46%)]; p = 0.02. Median AFP was also higher in subjects without HIV [1000 (IQR 83, 1000)] vs. those with HIV [505 (IQR 6,1000)]; p = 0.06 [Table 1].

### Overall survival outcomes

By the end of follow up, survival information was available for 191 subjects. Of those, 27/33 (82%) subjects with HIV and 119/158 (75%) subjects without HIV had died [HR = 1.55, 95%

**Table 1. Characteristics of HCC study participants by HIV status at study entry.**

| Parameter | Overall | HIV negative | HIV positive | P value |
|---|---|---|---|---|
| Total Participants (%) | 213 | 177 (83) | 36 (17) | |
| **Demographics** | | | | |
| Age,(yrs) median (IQR) | 52 (42, 60) | 52 (40, 60) | 51 (43, 56) | 0.74 |
| Males (n, %) | 150 (71) | 131 (74) | 19 (53) | 0.02 |
| BMI, median (kg/m2)(IQR) | 22 (20, 24) | 22 (20, 24) | 22 (20, 25) | 0.80 |
| **Highest level of education (n, %)** | | | | 0.79 |
| No formal education | 28 (13) | 25 (14) | 3 (8) | |
| Primary | 44 (21) | 36 (21) | 8 (22) | ' |
| Secondary | 51 (24) | 43 (25) | 8 (22) | |
| Tertiary (college and above) | 88 (42) | 71 (41) | 17 (47) | |
| Occupation, employed (n, %) | 202 (95) | 169 (96) | 33 (92) | 0.40 |
| **Marital status (n, %)** | | | | <0.001 |
| Single | 17 (8%) | 15 (9) | 2 (6) | |
| Married | 181 (85) | 157 (89) | 24 (67) | |
| Divorced | 1 (0.5) | 1 (0.6) | 0 (0) | |
| Widowed | 9 (4) | 3 (2) | 6 (17) | |
| Separated | 4 (2) | 0 (0) | 4 (11) | |
| **Clinical** | | | | |
| Family history HCC (n, %) | 16 (8) | 15 (9) | 1 (3) | 0.62 |
| Current alcohol use (n, %) | 45 (21) | 36 (20) | 9 (25) | 0.25 |
| Clinical symptoms ≥1 (n, %) | 177 (99%) | 147 (99) | 30 (100) | 1.0 |
| Advanced liver disease > = 1 symptom[‡] (n, %) | 162 (89%) | 137 (91%) | 25 (81%) | 0.12 |
| Child Pugh Turcotte Score (CPT) (n, %) | | | | 0.62 |
| A = 5–6 | 45 (22) | 35 (21) | 10 (29) | |
| B = 7–9 | 75 (37) | 63 (38) | 12 (34) | |
| C = 10–15 | 82 (41) | 69 (41) | 13 (37) | |
| **BCLC stage n (%)** | | | | 0.23 |
| Stage 0 | 1 (1) | 1 (1) | 0 (0) | |
| Stage A | 6 (4) | 6 (3) | 3 (10) | ' |
| Stage B | 38 (19) | 33 (20) | 5 (17) | |
| Stage C | 59 (30) | 51 (30) | 8 (28) | |
| Stage D | 94 (48) | 81 (48) | 13 (45) | |
| **Laboratory** | | | | |
| HBsAg positive (n, %) | 109 (52) | 91 (53) | 18 (50) | 0.86 |
| HBeAg (n,%) [†] | 8 (7) | 8 (9) | 0 (0) | 0.36 |
| Anti-HBe (n, %)[†] | 60 (55) | 52 (57) | 8 (44) | 0.31 |
| HBV DNA IU/mL median (IQR) | 12,850 (84–388,921) | 36,100 (336–606,750) | 47 (0, 461) | <0.001 |
| HBV DNA<10 IU/mL (n, %)[†] | 8 (7) | 3 (3) | 5 (28) | 0.005 |
| HBV active antivirals[†] | 47 (43%) | 38 (42%) | 9/18 (50%) | |
| Anti-HCV positive (n, %) | 75 (36) | 67 (39) | 8 (23) | 0.08 |
| HCV RNA IU/mL median (IQR) | 54,150 (0–470,750) | 21,800 (0–339,000) | 5,488,000 (179,015–1,032,500) | 0.17 |
| HCV RNA≥10 IU/mL (n, %) | 46 (61) | 40 (60) | 6 (75) | 0.66 |
| ALT (iu/l)(median, IQR) | 46 (25,75) | 45 (24, 75) | 50 (30, 77) | 0.46 |
| AST (iu/l)(median, IQR) | 159 (81, 324) | 162 (83, 321) | 129 (77, 462) | 0.95 |
| Albumin(g/l) (median, IQR) | 31 (26, 36) | 31 (26, 37) | 30 (27, 35) | 0.84 |
| Total bilirubin (mmol/l)(median, IQR) | 20 (7, 83) | 20 (8, 74) | 22 (6, 175) | 0.60 |
| INR (median, IQR) | 2 (1,3) | 2 (1,3) | 1 (1,2) | 0.19 |

*(Continued)*

**Table 1.** (Continued)

| Parameter | Overall | HIV negative | HIV positive | P value |
|---|---|---|---|---|
| Creatinineu(umol/l) (median, IQR) | 70 (55, 91) | 69 (54, 89) | 71 (58, 118) | 0.33 |
| Platelets(x10³/ul) (median, IQR) | 227 (167, 320) | 219 (160, 308) | 260 (191, 340) | 0.08 |
| AFP(ng)≥1000 vs.<1000 | 106 (54) | 91 (55) | 15 (48) | 0.56 |
| *HIV patients only* | | | | |
| On ART | - | - | 30 (83) | |
| ART treatment duration (median yrs, IQR) | - | - | 8 (2,12) | - |
| CD4 count cells/mm³ (median, IQR) | - | - | 284 (137, 356) | - |
| HIV VL <20 IU/L | - | | 15 (54) | - |
| **Radiologic** | | | | |
| Cirrhosis (n, %) | 121 (67) | 108 (71) | 13 (46) | 0.02 |
| Ascites (n, %) | 104 (58) | 91 (60) | 13 (45) | 0.29 |
| Varices (n, %) | 5 (3) | 4 (3) | 1 (4) | 0.16 |
| Number of lesions, (median, IQR) | 6 (3,11) | 7.0 (4,11) | 4 (1,11) | 0.16 |
| Liver mass(cm) diameter, (median, IQR) | 8 (5,12) | 8 (5,12) | 7 (6,13) | 0.89 |
| Tumor burden >50% | 88 (65) | 77 (66) | 11 (58) | 0.61 |
| Portal vein invasion, n (%) | 40 (23) | 37 (25) | 3 (11) | 0.16 |
| Portal vein thrombosis, n (%) | 34 (19) | 30 (20) | 4 (15) | 0.41 |

Abbreviations: na = not applicable; IQR = interquartile range; BCLC = Barcelona Clinic Liver Cancer; ART antiretroviral therapy; BMI body mass index; HCC hepatocellular carcinoma; AFP alpha fetoprotein

[†] denominator = all HBsAg positive patients

[‡] Symptoms of advanced liver disease ≥ 1: patient had at least one of the following symptoms marked yes i) Jaundice ii) abdominal swelling iii) irrational talk iv) Hematemesis v)Leg swelling vi) slow mentation vii) confusion

CI 1.02, 2.37, p = 0.04]. Median overall survival (OS) was 2.73 (95%CI 1.91, 3.94) months and was significantly shorter among subjects with HIV [HR = 0.99, 95%CI 0.59, 2.96] than those without HIV [HR = 3.02, 95%CI 2.1, 5.45]; p = 0.04 (Fig 1). Median survival declined with increasing BCLC stage (Fig 2). Among those in whom death data was available, the most

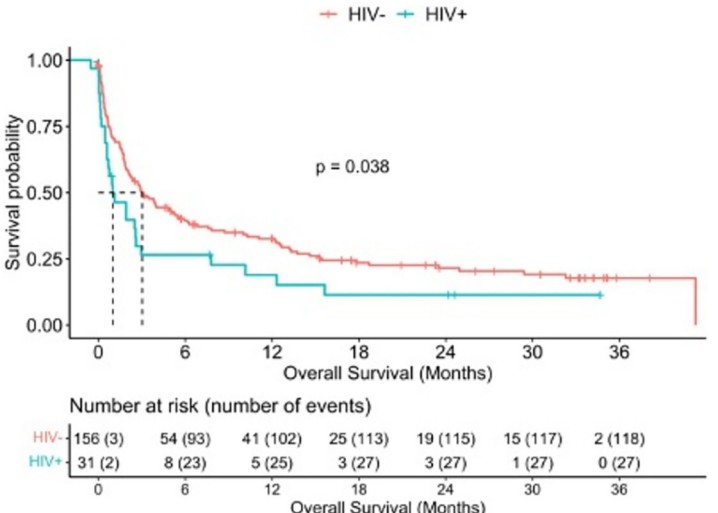

**Fig 1. Overall survival by HIV status.**

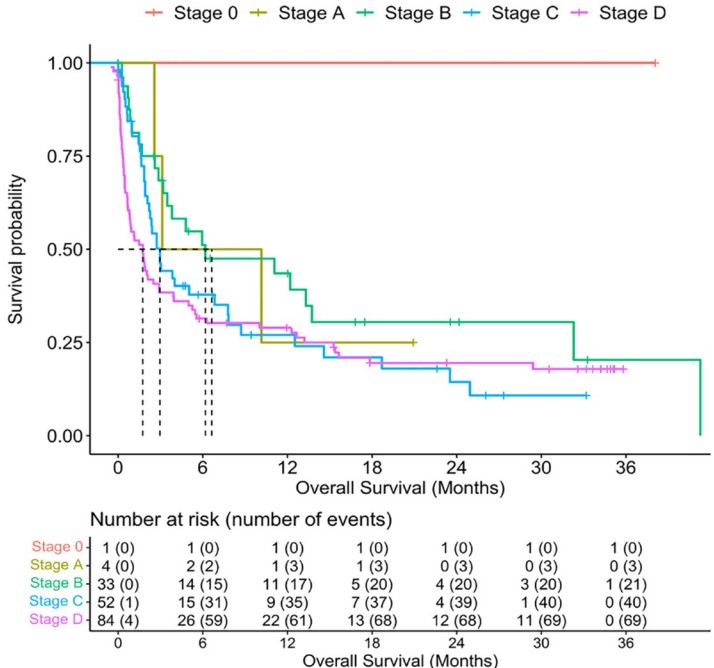

**Fig 2. Overall survival by BCLC.**

common cause of death was related to tumor progression or complications (98%). 8% died from variceal bleeding.

## Predictors of overall survival

Using the Cox proportional hazards model, HIV was found to be significantly associated with overall survival at the univariate level [HR = 1.55, 95%CI 1.02, 2.37, p = 0.04]. However, after adjusting for other well-known risk factors associated with mortality in persons with HCC (gender, current alcohol use, AFP, albumin, and total bilirubin), the association between HIV and survival was no longer significant [HR = 1.36, 95%CI 0.82, 2.24, p = 0.24]. Albumin [HR = 0.96, 95%CI 0.94, 0.99; p = 0.003] and AFP level ≥1000ng/mL [HR = 1.63, 95%CI 1.81,2.3; p = <0.01] were the only significant predictors of survival, holding all other variables constant. HCV was not associated with overall survival [Table 2].

## Discussion

In this study, comparing host and tumor characteristics and prognosis in Nigerian adults with HCC with and without HIV, we found most individuals presented with symptomatic late-stage disease. Despite few clinical, or radiologic differences at presentation between the two groups, persons with HIV had a significantly shorter median survival compared to those without HIV, although this was not significant after adjusting for other HCC mortality risk factors. Other independent predictors of mortality included high AFP and low albumin levels, which have been reported previously in association with advanced HCC and HCC-related death [20, 21].

Our data adds to a growing number of studies reporting an adverse effect of HIV on HCC outcomes, several of which have reported similarly reduced survival among PLH and an independent effect of HIV on mortality [14, 20–33]. The mechanisms underlying the effects of

**Table 2. Predictors associated with overall survival.**

| Predictor | Univariate HR (CI) | P value | Multivariate HR (CI) | P value |
|---|---|---|---|---|
| Age | 1.01 (0.998, 1.02) | 0.10 | - | - |
| Sex (Male vs. Female) | 1.23 (0.84, 1.79) | 0.28 | 1.43 (0.94, 2.15) | 0.10 |
| Current alcohol use (Yes vs. No) | 1.03 (0.7, 1.51) | 0.89 | 0.66 (0.43, 1.03) | 0.07 |
| HBsAg status (Positive vs. Negative) | 0.82 (0.59, 1.13) | 0.23 | - | - |
| HCV RNA | 1.00 (1.00, 1.00) | 0.46 | - | - |
| AFP(ng) ≥1000 vs. <1000 | 1.66 (1.81, 2.33) | <0.01 | 1.63 (1.15, 2.3) | <0.01 |
| Child Pugh Turcotte Score | | | | |
| B vs A | 1.10 (0.68, 1.71) | 0.75 | - | - |
| C vs A | 1.49 (0.95, 2.33) | 0.08 | - | - |
| Albumin | 0.97 (0.95, 0.99) | 0.01 | 0.96 (0.94,0.99) | <0.01 |
| Total bilirubin | 1.00 (1, 1.00) | 0.02 | 1.00 (1.00,1.00) | 0.16 |
| HIV (Positive vs. Negative | 1.55 (1.02, 2.37) | 0.04 | 1.38 (0.84, 2.29) | 0.24 |
| Total tumor burden >50% vs. <50% | 1.41 (0.92, 2.18) | 0.12 | - | - |

Note: Univariable analyses and multivariable analyses (including Gender, Alcohol Use, AFP, Albumin, Total Bilirubin, HIV status) were performed using the Cox proportional hazards model. Unadjusted p-values were calculated using Cox's regression. Hazard ratio (HR) and corresponding 95% confidence interval (CI) are presented.

HIV on HCC are not clear. Some have suggested the poor prognosis observed in those with HIV may be due to the long-term effects of HIV and its associated immune dysregulation on tumor biology, causing tumors to be more aggressive [10]. Indeed, a few studies have reported a higher prevalence of more aggressive features in PLH such as infiltrative tumors and portal invasion, however, most of these have been in persons co-infected with HCV [23]. In our study, we did not distinguish between infiltrative vs. other tumor types. However, tumor burden >50% and portal vein invasion was similar in subjects with and without HIV.

Others have suggested that more limited access to cancer therapies and treatment ineligibility may account for the poorer prognosis among those with HIV and HCC [20]. Interestingly, improved access to therapies among PLH did not translate into better prognostic outcomes in two large studies where a higher mortality was observed in PLH [22, 24]. Treatment intent also did not differ between persons with and without HIV in a large VA study where PLH had 37% higher risk of death [25]. In our study, none of the participants received any HCC specific treatments, thus access to treatment could not have accounted for the observed mortality differences.

Whether stage of HIV disease and HIV viremia at the time of diagnosis contributes to the earlier mortality observed in PLH remains debated. In one of the larger HCC outcome studies where a difference in mortality (higher in PLH) was observed, there was no independent association between CD4+ T cell count and death in PLH. However, median survival was longer in those with undetectable HIV RNA levels compared to those with viremia [22]. In another study where no differences in median survival between persons with HCC with and without HIV were observed, the proportion of patients with HIV virologic suppression and CD4>200 was 87%, suggesting that virologic suppression could have improved outcomes in PLH [27]. Although over 80% of PLH in our study were on ART treatment, not all were virologically suppressed. Unfortunately, due to small numbers, we were unable to determine the effect of virologic non-suppression on death in this cohort.

In both cohorts, regardless of HIV status, the prognosis was extremely poor and well over two-thirds (69%) died within the year. Overall survival was much shorter compared to other cohorts in Europe and North Africa [22, 23], likely because most participants presented with late-stage disease and no treatments, even supportive, were available. The WHO, AASLD and several other professional societies, currently recommend surveillance using semiannual abdominal ultrasonography in high-risk patients which includes patients with cirrhosis and other select groups, including Africans with chronic HBV infection at any age [15]. Ultrasound screening for HCC has been shown to improve early tumor detection, receipt of curative treatment, and overall survival in at-risk patients [10]. However, screening for HCC is rarely practiced as ultrasound is seldom available outside of academic settings and provider knowledge about HCC surveillance practices is poor [34].

In addition to the observed effects of HIV on HCC mortality in this study, another notable finding in both those with and without HIV, was the high prevalence (22%) of chronic HCV co-infection (defined as anti-HCV positive with HCV VL>10 IU/mL). Our study is one of the first to confirm active HCV infection with molecular testing in West Africa in persons with HCC. The prevalence of HCV among persons in this study far exceeds the national prevalence in the general population (1.73%) [3], suggesting HCV is an important risk factor for HCC in this region. In Europe and North America, HCV contributes to 30–60% cases of HCC countries [35–37] and is a leading indication for liver transplant in persons with HIV [38]. Until now, HCV has not been recognized as a major risk factor in West Africa primarily because of a lack of available confirmatory testing for HCV and limited knowledge of HCV among healthcare providers and patients. Our results highlight the urgent need for scale up of HCV screening in persons with and without HIV in Nigeria and HCC surveillance in these high-risk individuals, as well as improved access to HCV curative therapies.

HBV was also highly prevalent in this cohort (52%), a finding not unexpected since Nigeria is a region that is considered hyperendemic for HBV, with high rates of infection in those with and without HIV (9.9%-13.2%) [39]. Previous studies have found HBV in over 60% of persons with HCC in Nigeria, similar to our data [11]. In studies from Asia, both high HBV serum DNA levels and HBeAg seropositivity have been shown to correlate with more advanced HCC disease (portal vein tumor thrombosis and extrahepatic metastases) as well as a higher mortality [40]. Our cohort had very low prevalence of HBeAg seropositivity and low median HBV viral loads, most likely due to the high prevalence of circulating HBV genotype E strains, which we and others have previously reported [41–43]. Of note, only 42% of persons with HBV were on antivirals at the time of diagnosis and 76 (70%) did not meet criteria for antiviral initiation according to AASLD guidelines [44], at the time of HCC diagnosis. Cirrhosis was absent in a third of all patients, as seen in other cohorts [45], and less common in those with HIV. The relatively high prevalence of HCC occurring in non-cirrhotic livers and young age of onset, provides more evidence of hepatocarcinogensis at earlier stages of liver disease and the need for earlier and more intensive surveillance in this setting. The high proportion of persons with HBV mono-infection and HCC who were not eligible for treatment according to guidelines, also suggests antiviral therapy should also be considered sooner in these patients or given universally to all positive HBV patients. Whether antivirals are protective against HCC arising in non-cirrhotic livers and persons with less active HBV infection, however, is still to be determined.

## Limitations

There are some limitations to this study. One is the relatively small number of HIV cases (although still larger than in other African cohorts), that limits interpretation of findings. One

possible explanation for the lower numbers is the shorter survival of persons with HIV, especially those not yet on ART, which would prevent late complications such as HCC from being detected before their demise. Our cohort overall, presented in late-stage HCC and had few participants with early-stage disease reflecting the lack of surveillance for HCC. Due to a lack of resources and high costs of testing, most participants with HCC could not complete a full staging work up including a chest and abdominopelvic CT to evaluate for extrahepatic metastases or bone scan to evaluate for bone metastases. As a result, BCLC staging could have been underestimated. The strengths of the study include the use of CT imaging to confirm the diagnosis of HCC, which is recommended by AASLD, and data on several other host, infection, and prognostic markers.

## Conclusion

In conclusion, in one of the largest longitudinal cohort studies of HCC from West Africa to date, we showed that HIV was associated with shorter time to death and increased risk of mortality, although this was not significant after adjusting for other known risk factors including albumin and AFP. More aggressive HCC surveillance, diagnosis and treatment are desperately needed in Nigeria to improve the prognosis of this devastating disease, particularly in those infected with HIV, HBV and HCV. Special attention should be paid to improving access to treatment for HCV and HBV which account for a substantial burden of HCC in this region.

## Acknowledgments

We are grateful to the staff and clients that participated in this study. The authors acknowledge the following study team members for their contributions: Patience Omaiye, Atta Okute; Makupu Jire; Mark Henry; Edwin Adoga.

This study was previously presented in part at the 9th Annual Symposium on Global Cancer Research March 10–11, 2021. The abstract was also selected for publication in AACR's Cancer Epidemiology, Biomarkers, & Prevention special edition.

## Author Contributions

**Conceptualization:** Edith Okeke, Folasade T. Ogunsola, Lewis R. Roberts, Lifang Hou, Robert L. Murphy, Claudia A. Hawkins.

**Data curation:** Folasade T. Ogunsola.

**Formal analysis:** Kristen Bell, Kwang-Youn Kim.

**Funding acquisition:** Lifang Hou, Robert L. Murphy.

**Investigation:** Alani S. Akanmu, Claudia A. Hawkins.

**Methodology:** Pantong M. Davwar, Kwang-Youn Kim, Claudia A. Hawkins.

**Project administration:** Pantong M. Davwar, Mary Duguru, David Nyam, Emuobor A. Odeghe, Revika Singh, Godwin Imade, Alani S. Akanmu, Claudia A. Hawkins.

**Resources:** David Nyam, Emuobor A. Odeghe, Revika Singh, Godwin Imade, Alani S. Akanmu.

**Supervision:** Edith Okeke, Mary Duguru, David Nyam, Emuobor A. Odeghe, Ganiat Oyeleke, Olufunmilayo A. Lesi, Godwin Imade, Claudia A. Hawkins.

**Validation:** Olufunmilayo A. Lesi, Marion G. Peters, Lewis R. Roberts, Claudia A. Hawkins.

**Visualization:** Olufunmilayo A. Lesi, Marion G. Peters, Lewis R. Roberts, Claudia A. Hawkins.

**Writing – original draft:** Pantong M. Davwar, Edith Okeke, Revika Singh, Claudia A. Hawkins.

**Writing – review & editing:** Pantong M. Davwar, Edith Okeke, Mary Duguru, David Nyam, Kristen Bell, Emuobor A. Odeghe, Ganiat Oyeleke, Olufunmilayo A. Lesi, Atiene S. Sagay, Folasade T. Ogunsola, Marion G. Peters, Lewis R. Roberts, Claudia A. Hawkins.

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
