## [Decision Letter · Decision Letter 0]

16 Nov 2022

PONE-D-22-28133Hepatocellular carcinoma presentation and prognosis among Nigerian adults with and without HIV.PLOS ONE

Dear Dr. Hawkins,

Thank you for submitting your manuscript to PLOS ONE. After careful consideration, we feel that it has merit but does not fully meet PLOS ONE’s publication criteria as it currently stands. Therefore, we invite you to submit a revised version of the manuscript that fully addresses the points raised by the reviewers.

We look forward to receiving your revised manuscript.

Kind regards,

Matias A Avila, Ph.D.

Academic Editor

PLOS ONE

Journal Requirements:

"Research reported in this publication was supported by the National Cancer Institute of the National Institutes of Health under award number U54CA221205. The content is solely the responsibility of the authors and does not necessarily represent the official views of the National Institutes of Health."

"Grant Number: NIH/NCI U54CA221205 (PI Robert Murphy, Lifting Hou)

5. Please amend the manuscript submission data (via Edit Submission) to include author "Atiene Sagay".

6. Please amend your list of authors on the manuscript to ensure that each author is linked to an affiliation. Authors’ affiliations should reflect the institution where the work was done (if authors moved subsequently, you can also list the new affiliation stating “current affiliation:….” as necessary).

7. We note that you have included the phrase “data not shown” in your manuscript. Unfortunately, this does not meet our data sharing requirements. PLOS does not permit references to inaccessible data. We require that authors provide all relevant data within the paper, Supporting Information files, or in an acceptable, public repository. Please add a citation to support this phrase or upload the data that corresponds with these findings to a stable repository (such as Figshare or Dryad) and provide and URLs, DOIs, or accession numbers that may be used to access these data. Or, if the data are not a core part of the research being presented in your study, we ask that you remove the phrase that refers to these data.

8. Please include a separate caption for each figure in your manuscript.

9. We note you have included a table to which you do not refer in the text of your manuscript. Please ensure that you refer to Table 2 in your text; if accepted, production will need this reference to link the reader to the Table.

Reviewers' comments:

Reviewer's Responses to Questions

**Comments to the Author**

1. Is the manuscript technically sound, and do the data support the conclusions?

Reviewer #1: Yes

Reviewer #2: Yes

2. Has the statistical analysis been performed appropriately and rigorously? 

Reviewer #1: No

Reviewer #2: Yes

3. Have the authors made all data underlying the findings in their manuscript fully available?

Reviewer #1: Yes

Reviewer #2: Yes

4. Is the manuscript presented in an intelligible fashion and written in standard English?

Reviewer #1: Yes

Reviewer #2: Yes

5. Review Comments to the Author

Reviewer #1: Dear colleagues,

This study compares patients with hepatocellular carcinoma with or without concomitant Human Immunodeficiency Virus. 213 patients were prospectively enrolled, and clinical characteristics and survival is reported. The first aim is to compare the survival between groups. Other aims are to compare different clinical characteristics.

Main comments:

1. The multivariate survival analysis shows us the main endpoint, and it should be the main conclusion. BCLC should be included, as you show in figure 2 as related with survival. AFP cut at 1000 seems to be confusing, as it is the highest value reported. We don’t know the cut value of albumin. Altogether could change the results. Consider pointing out in the methods and abstract that this is the primary endpoint.

2. Demographics about late-stage diagnosis, poor access to curative treatments, coinfection with HCV and HBV, as well as other comorbidities are very important information that must be reported. It seems that HIV+ or HIV- HCC could have different behavior or may have different relationship with healthcare. For instance: HIV- are diagnosed more frequently with ChildPugh score C, and HIV+ are more frequently diagnosed in BCLC A stage and less cirrhotic.

Minor comments:

1. Line 17: cirrhosis is more frequent in HIV-. Following the same idea, it seems that there is more child-pugh C. Please note that BCLC-D include child-pugh C score patients, so consider reviewing only 24 HIV negative patients are BCLC-D when 34 HIV negative patients are child pugh score C.

2. Line 64: exlcusion criteria include other malignancies and they are usually related to HIV, so it may create a confuse factor.

3. Line 81-82: For me it’s not clear if patients enrolled are considered for sorafenib or not, and the reason. I think the access to systemic treatment is a key point in this report.

4. Line 121: please report the data and reference tables and figures.

I think this is a very interesting report. It is pointing out the situation of HCC in Nigeria, which is too different of US or European Countries. I think it is extremely important to publish this data in a good journal, so it is a well conducted prospective cohort study. May be it can improve the statistical analysis, but it would be very interesting for all of the people treating HCC to know the behavior in Nigeria.

Reviewer #2: In this study, authors present a descriptive analysis of HCC in Nigerian population. Describes demographics, tumor characteristics, Virus B or C infection, liver function, and HIV infection, and compares characteristics and overall survival between HIV positive and HIV negative patients. The main result of this study is that the majority of patients (70%) are diagnosed at intermediate or advance stage, any patient could received treatment for the HCC and the survival was very dissapointing.

Major issues: HCC diagnosis is based on AASLD guidelines, that states the diagnosis of HCC based on radiological enhancement characteristics. However, this radiological diagnosis " cannot be made by imaging in patients without cirrhosis, even if enhancement and washout are present, and biopsy is required in these cases". In the present study, in one third of the patients, cirrhosis was absent, but no biopsy was performed. This could provide a misdiagnosis in some cases, and liver nodules included as HCC could correspond to othere lesions, such as intrahepatic cholangiocarcinoma or others. This remains an important issue when analysing survival.

Minor issues. Even though Sorafenib was offered to eligible patitns with advanced disease when available, none patients received sorafenib. Which was the reason?

Units for variables are missing along the text and in the tables. Above all, it remains necessary in tumor diameter, lesions are sized in cm or in mm?

In the figures, legends are referred as Arm 1 HCC+patients, suggesting as if there were and Arm 2. It seems like coming from other analysis....

6. PLOS authors have the option to publish the peer review history of their article (what does this mean?). If published, this will include your full peer review and any attached files.

Reviewer #1: **Yes: **Carles Fabregat-Franco

Reviewer #2: No

---

## [Author Response · Author response to Decision Letter 0]

14 Feb 2023

Reviewer 1 Main Comments 

1. The multivariate survival analysis shows us the main endpoint, and it should be the main conclusion. BCLC should be included, as you show in figure 2 as related with survival. 

Response: We thank the reviewer for this important point. We revised the conclusions to state the main findings of the multivariate analysis more clearly. BCLC was not included in the multivariate analysis because the proportional hazards assumption was not met with this variable. This was also true with the updated BCLC data.

2. AFP cut at 1000 seems to be confusing, as it is the highest value reported. We don’t know the cut value of albumin. Altogether could change the results. Consider pointing out in the methods and abstract that this is the primary endpoint. 

Response: Thank you for this comment. The table reports AFP as median and IQR, therefore 1000 is not the highest value reported. We revised table 1 to report AFP as a categorical rather than continuous value (>1000 vs. 1000) to avoid any confusion. 54% of the total cohort had AFP levels >1000. AFP was not the primary outcome of this study. OS was the primary outcome of this study which we have better clarified at line 113.

3. Demographics about late-stage diagnosis, poor access to curative treatments, coinfection with HCV and HBV, as well as other comorbidities are very important information that must be reported. It seems that HIV+ or HIV- HCC could have different behavior or may have different relationship with healthcare. For instance: HIV- are diagnosed more frequently with ChildPugh score C, and HIV+ are more frequently diagnosed in BCLC A stage and less cirrhotic. 

Response: We agree with these comments. We were able to collect comprehensive data on clinical and radiologic characteristics that could be used to determine stage of disease as well as other co-morbidities that are known to be associated with liver cancer such as family history and viral hepatitis B and C co-infection. None of our study participants had access to oral chemotherapies (sorafenib) or interventional (surgical or non-surgical) procedures. The proportions in each CPT class/BCLC stage did not differ significantly between those with and without HIV. (Table 1). Thus, even a different relationship with healthcare, which was not measured but presumed to be better among persons with HIV, was unlikely to have improved the likelihood of getting an intervention or being diagnosed earlier.

Reviewer 1 Minor Comments 

1. Line 17: cirrhosis is more frequent in HIV-. Following the same idea, it seems that there is more child-pugh C. Please note that BCLC-D include child-pugh C score patients, so consider reviewing only 24 HIV negative patients are BCLC-D when 34 HIV negative patients are child pugh score C.

Response: We appreciate the reviewer bringing this discrepancy to our attention. We went back and reviewed all of our CPT and BCLC data and found that some of the BCLC and CPT scores had been incorrectly calculated. These have now been corrected. Additional scores could also be calculated for persons who had missing data at the time of the previous analysis. (Table 1.)

2. Line 64: exlcusion criteria include other malignancies and they are usually related to HIV, so it may create a confuse factor. 

Response: Exclusion criteria for this study included any current or past malignancy whether related to HIV or not. We further clarified this at line 64. We are not sure what was meant by ‘create a confuse factor’ but would be happy to respond to this once clarified.

3. Line 81-82: For me it’s not clear if patients enrolled are considered for sorafenib or not, and the reason. I think the access to systemic treatment is a key point in this report. 

Response: Thank you for this comment. Unfortunately, none of the participants were able to access sorafenib in this study due to unavailability or high costs of the medication. We further clarified this in lines 83-84.

4. Line 121: please report the data and reference tables and figures.

Response:Table 1 is referenced at line 138 at the end of the first paragraph which summarizes all the data in this table. Figures 1 and 2 are referenced at lines 142 and 144 respectively.

Reviewer 2 Main Comments 

1. HCC diagnosis is based on AASLD guidelines, that states the diagnosis of HCC based on radiological enhancement characteristics. However, this radiological diagnosis " cannot be made by imaging in patients without cirrhosis, even if enhancement and washout are present, and biopsy is required in these cases". In the present study, in one third of the patients, cirrhosis was absent, but no biopsy was performed. This could provide a misdiagnosis in some cases, and liver nodules included as HCC could correspond to othere lesions, such as intrahepatic cholangiocarcinoma or others. This remains an important issue when analysing survival.

Response: Thank you for this comment and we acknowledge this is an area where there is some disagreement among experts. AASLD guidelines that were used for this study are considered equivalent to those used in LI-RADS 5. LI-RADs is designed for use in livers at risk for developing HCC including cirrhosis and viral infections like HBV without cirrhosis. Liver biopsies are not required for confirmation of diagnosis in LI-RADs 5. Our study population was one that was very high risk with >50% HBsAg seropositive and 22% with active HCV infection, therefore we believe that the risk of misdiagnosis when applying this criterion was extremely low. 

Reviewer 2 Minor Comments 

1. Even though Sorafenib was offered to eligible patitns with advanced disease when available, none patients received sorafenib. Which was the reason? 

Response: Unfortunately, none of the participants were able to access sorafenib in this study due to unavailability or high costs of the medication. We further clarified this in lines 83-84.

2. Units for variables are missing along the text and in the tables. Above all, it remains necessary in tumor diameter, lesions are sized in cm or in mm? 

Response: Thank you for this comment. We have added units and measurements to Tables 1 and 2.

3. In the figures, legends are referred as Arm 1 HCC+patients, suggesting as if there were and Arm 2. It seems like coming from other analysis....

Response: The reviewers are correct, HCC patients in this analysis were participants enrolled in a larger study of persons with HIV with (Arm 1) and without (Arms 2, 3) HCC. We have revised the legend to make this less confusing.

---

## [Editor Report · Decision Letter 1]

17 Feb 2023

Hepatocellular carcinoma presentation and prognosis among Nigerian adults with and without HIV.

PONE-D-22-28133R1

Dear Dr. Hawkins,

We’re pleased to inform you that your manuscript has been judged scientifically suitable for publication and will be formally accepted for publication once it meets all outstanding technical requirements.

Kind regards,

Matias A Avila, Ph.D.

Academic Editor

PLOS ONE
---

## [Editor Report · Acceptance letter]

24 Feb 2023

PONE-D-22-28133R1 

Hepatocellular carcinoma presentation and prognosis among Nigerian adults with and without HIV. 

Dear Dr. Hawkins:

I'm pleased to inform you that your manuscript has been deemed suitable for publication in PLOS ONE. Congratulations! Your manuscript is now with our production department. 

Kind regards, 

on behalf of

Dr Matias A Avila 

Academic Editor

PLOS ONE